
# Towards a nappe theory: Thermo-mechanical simulations of nappe detachment, transport and stacking in the Helvetic Nappe System, Switzerland

Dániel Kiss[1], Thibault Duretz[2,1], and Stefan M. Schmalholz[1]

[1]Institute of Earth Sciences, University of Lausanne, 1015 Lausanne, Switzerland
[2]Univ. Rennes, CNRS, Géosciences Rennes - UMR 6118, F-35000 Rennes, France

**Correspondence:** Dániel Kiss (daniel.kiss@unil.ch)

**Abstract.**

Tectonic nappes are observed for more than a hundred years. Although geological studies often refer to a "nappe theory", the physical mechanisms of nappe formation are still incompletely understood. We apply two-dimensional numerical simulations of shortening of a passive margin, to investigate the thermo-mechanical processes of detachment, transport and stacking of
nappes. We use a visco-elasto-plastic model with standard creep flow laws and Drucker-Prager yield criterion. We consider tectonic inheritance with two initial mechanical heterogeneities: (1) lateral heterogeneity of the basement-cover interface due to half-grabens and horsts and (2) vertical heterogeneities due to layering of mechanically strong and weak sedimentary units. The model shows detachment and horizontal transport of a thrust nappe and stacking of this thrust nappe above a fold nappe. The detachment of the thrust sheet is triggered by stress concentrations around the sediment-basement contact and the resulting
brittle-plastic shear band formation. The horizontal transport is facilitated by a basal shear zone just above the basement-cover contact, composed of thin, weak sediments. Fold nappe formation occurs by a dominantly ductile closure of a half-graben and the associated extrusion of the half-graben fill. We apply our model to the Helvetic nappe system in Western Switzerland, which is characterized by stacking of the Wildhorn thrust nappe above the Morcles fold nappe. The modeled structures and temperature field agree with data from the Helvetic nappe system. The mechanical heterogeneities must generate contrasts in
effective viscosity (i.e. ratio of stress to strain rate) of four orders of magnitude to model nappe structures similar to the ones of the Helvetic nappe system.

# 1   Introduction

Tectonic nappes were discovered more than a hundred years ago and are considered as typical tectonic features of orogenic
belts (e.g. Price and McClay, 1981), particularly in the Alps (e.g. Lugeon, 1902; Termier, 1906; Argand, 1916; Tollmann, 1973; Trümpy, 1980; Escher et al., 1993; Pfiffner, 2014). Several definitions of a nappe have been proposed (see discussion in Price





and McClay, 1981), for example, a basic definition modified after Termier (1922) is: "A nappe is a rock packet not in its place, resting on a substratum that is not its original one". Two end-member types of nappes are commonly distinguished, namely fold nappes and thrust nappes, or thrust sheets (e.g. Termier, 1906; Price and McClay, 1981; Epard and Escher, 1996). Fold nappes are recumbent folds exhibiting large-scale stratigraphic inversion, typically with amplitudes that are exceeding several

kilometers. In contrast, thrust sheets are allochtonous sheets with a prominent shear zone or thrust at their base, but without a prominent overturned limb. The importance of tectonic nappes for orogeny, especially for collisional orogens, is nowadays well established and many geological studies refer to a "nappe theory" when mentioning tectonic nappes. However, the physical mechanisms of nappe detachment, transport and stacking are still incompletely understood.

We focus here on the Helvetic nappe system in Western Switzerland (see next section for a geological overview), which is

one of the birthplaces of the concept of tectonic nappes. Arnold Escher mentioned a nappe (he used "Decke" in german) and a colossal overthrust ("colossale Überschiebung") in 1841 during the presentation of a geological map of the canton Glarus, Eastern Switzerland (Escher von der Linth, 1841). Escher did not dare to publish his interpretation, but explained it in the field to Roderick Murchinson, who published the overthrust interpretation in 1849 (Murchison, 1849), crediting Escher for the original observation. Bertrand (1884) argued also convincingly for an overthrust nappe (he used "masse de recouvrement" and

"lambeaux de recouvrement" instead of nappe) in the Glarus region so that finally also Heim (1906) accepted the overthrust interpretation instead of the earlier preferred double-fold interpretation ("Überschiebungsfalte" instead of "Doppelfalte"). Although the important controversies and observations supporting tectonic nappes are related to the Glarus region, which is part of the Helvetic nappe system, the true birth date of the nappe concept in the Alps, according to Trümpy (1991), is the publication by Schardt (1893) who worked in the Prealps, belonging to the Penninic domain (e.g. Escher et al., 1993). Schardt (1893)

realized that Jurassic breccias have been thrust over Tertiary flysch and that large regions of the Prealps have been actually emplaced as a major overthrust. After decades of controversy, the existence of nappes got generally accepted approximately a century ago, revolutionizing Tectonics, Alpine Geology and orogeny in general (for historical reviews see Bailey 1935; Masson 1976; Merle 1998; Trümpy 1991; Dal Piaz 2001; Schaer 2010).

Since then, a considerable effort has been made in mapping the present-day structure of the Helvetic nappe system (e.g.

Steck, 1999; Pfiffner et al., 2011). Structural and paleogeographic reconstructions have provided a valuable insight into the kinematics of nappe formation (e.g. Gillcrist et al., 1987; Epard and Escher, 1996; Herwegh and Pfiffner, 2005; Bellahsen et al., 2012; Boutoux et al., 2014). Therefore, the geometrical structure and kinematic evolution of the Helvetic nappe system is reasonably well understood. There are also theoretical and analogue modeling studies investigating the formation of fold-and-thrust belts and nappes (e.g. Bucher, 1956; Rubey and King Hubbert, 1959; Dietrich and Casey, 1989; Merle, 1989; Casey

and Dietrich, 1997; Wissing and Pfiffner, 2003; Bauville et al., 2013; Poulet et al., 2014; Erdős et al., 2014; Jaquet et al., 2014; Ruh et al., 2014; Bauville and Schmalholz, 2017). However, the controlling physical processes of nappe detachment, transport and stacking, and the associated dominant rock deformation mechanism are still incompletely understood and, therefore, the frequently mentioned "nappe theory" is not complete. For example, for fold nappes, many interpretations favor distributed shearing and dominantly ductile deformation mechanisms, such as dislocation or grain-size sensitive diffusion creep (e.g. Ram-

say et al., 1983; Gillcrist et al., 1987; Ebert et al., 2008; Bauville et al., 2013). However, there are also interpretations arguing





for localized thrusting and dominantly brittle-plastic deformation mechanisms, such as fracturing related to fluid pressure (e.g. Boyer and Elliott, 1982; Granado and Ruh, 2019). Furthermore, the presumed driving forces of nappe transport are either external surface forces, due to tectonic compression, or internal body forces, due to gravity. Heterogeneous shearing due to buttressing in a general compressional regime is an example of deformation driven by external forces (e.g. Ramsay et al., 1983;

Epard, 1990; Bauville et al., 2013; Boutoux et al., 2014). Gravity gliding and spreading is an example of deformation driven by body forces (e.g. Durney, 1982; Merle, 1989; Merle and Guillier, 1989).

For thrust sheets, the variety of the proposed emplacement mechanisms is even larger than for fold nappes (see for an overview Merle, 1998). The apparently straightforward interpretation that thrust sheet transport is controlled by frictional sliding is problematic, because stresses required to move a long sheet over a frictional surface exceed the strength of the rock

sheet so that the sheet would break into smaller pieces (e.g. Smoluchowski, 1909; King Hubbert and Rubey, 1959; Price and Cosgrove, 1990). This problem is known as the overthrust paradox (e.g. Smoluchowski, 1909; King Hubbert and Rubey, 1959). Several solutions for this paradox were proposed, such as (1) reduction of the effective stress due to pore fluid pressure causing a reduction of the effective friction angle (e.g. King Hubbert and Rubey, 1959; Rubey and King Hubbert, 1959) or (2) a dominantly ductile deformation mechanism (e.g. Smoluchowski, 1909; Goguel, 1948; Voight, 1976), presumably in

combination with thermally-, chemically- or mechanically-activated softening mechanisms (e.g. Poirier, 1980; Ebert et al., 2008; Poulet et al., 2014). Another problem of purely brittle-frictional interpretations, assuming homogeneous material properties, is that thrust sheets have often been displaced over tens of kilometers on sub-horizontal thrust planes or shear zones. However, according to Anderson's theory of faulting (e.g. Turcotte and Schubert, 2014) thrust planes for friction angles of ca 30 degrees should dip with ca 30 degrees with respect to the horizontal, if the smallest principal stress, $\sigma_3$, is approximately

vertical. Smaller friction angles would increase the dip angle. For example, for zero friction angle, corresponding to a constant, pressure-insensitive yield stress, the dip angle would be 45 degrees. Therefore, prominent low-angle thrust planes are likely controlled by mechanical heterogeneities, such as the orientation of the basement-cover interface and of mechanically weak shale-rich or evaporite layers, as has been suggested for the Hevetic nappe system (e.g. Pfiffner, 1993; Steck, 1999; Pfiffner et al., 2011; Bauville and Schmalholz, 2017).

To make another step towards a "nappe theory" that explains the physical process of nappe formation, we investigate the detachment, transport and stacking of nappes with two-dimensional (2D) numerical simulations based on continuum mechanics. To keep the model relatively simple and transparent, we focus here on thermo-mechanical processes on the macro-scale, larger than the typical size of mineral grains. Hence, we do not consider hydro-chemical couplings, such as fluid release by carbonate decomposition (e.g. Poulet et al., 2014), and micro-scale processes, such as micro-structural evolution with secondary

phases (e.g. Herwegh et al., 2011). The numerical algorithm is based on the finite difference method. We consider a standard visco-elasto-plastic deformation behavior, heat transfer and thermo-mechanical coupling by shear heating and temperature-dependent viscosities. We also apply velocity boundary conditions that are standard for modeling accretionary or orogenic wedges (e.g. Buiter et al., 2006). For the comparison between model results and natural observations, we consider a geological section across the Helvetic Nappe System in Western Switzerland. This section is characterized by two deformed basement

massifs, the Aiguilles-Rouges and Mont-Blanc massifs, a fold nappe, the Morcles nappe, and a thrust nappe, the Wildhorn





super-nappe, that has been overthrust, or stacked, above the underlying fold nappe (Fig. 1). In our models, we consider the tectonic inheritance of the Mesozoic passive margin formation in the form of simple half-grabens and horsts, because the Helvetic nappe system resulted from the inversion of the pre-Alpine European passive margin (e.g. Trümpy, 1980). We consider two main orientations of inherited mechanical heterogeneities: (1) a lateral variation of mechanical strength due to the lateral

alternation of basement and sediments associated with the half-graben structure and (2) a vertical variation of strength due to (i) the basement-cover interface, (ii) the alternation of strong carbonate with weak shale-rich units (so-called mechanical stratigraphy after Pfiffner (1993)) and (iii) the pressure and temperature sensitivity of rock strength and effective viscosity, respectively.

The main aim of this study is to show that a thermo-mechanical model based on the theory of continuum mechanics (i) with

a well established visco-elasto-plastic deformation behaviour using standard flow laws, (ii) with mechanical heterogeneities mimicking pre-Alpine extensional heritage and stratigraphic layering and (iii) with a wedge-type compressional configuration can self-consistently explain the first-order features of nappe detachment, transport and stacking in the Helvetic nappe system.

## 2 Short overview of the Helvetic Nappe System in Western Switzerland

The Helvetic nappe system is commonly subdivided into Infrahelvetic, Helvetic and Ultrahelvetic units (Fig. 1c) (e.g. Masson

et al., 1980; Escher et al., 1993; Pfiffner et al., 2011). The nappes consist mainly of Jurrasic to Paleogen sediments that were deposited on the Mesozoic European passive margin before the Alpine orogeny (Fig. 1a). This passive margin inherited half-grabens and horsts from the Mesozoic, pre-Alpine extensional phase (e.g. Masson et al., 1980; Escher et al., 1993). The stratigraphy of the nappes is generally characterized by shale-rich units, totaling several kilometers in thickness, and two major units of massive platform carbonates, the so-called Quinten (Malm) and Urgonian (Lower Cretaceous) limestones, with a

thickness of several hundred meters (e.g. Masson et al., 1980; Pfiffner, 1993; Pfiffner et al., 2011).

In the studied section, along the Rhone-valley near Martigny (Switzerland), the Infrahelvetic units form the Morcles fold nappe (e.g. Steck, 1999). This recumbent fold nappe is strongly deformed, but is still connected to its original position of deposition, the Mesozoic half-graben between the Aiguilles-Rouges and the Mont-Blanc massifs (Fig. 1a). Therefore, the Morcles nappe is considered as a parautochtonous unit and its root zone, between the Aiguilles-Rouges and the Mont-Blanc

massifs, is termed the Chamonix zone (Fig. 1c). The sediments forming the Helvetic nappes have been deposited on more distal regions of the European passive margin than the units forming the Morcles nappe. The original regions of deposition of the Infrahelvetic and the Helvetic units have been separated by a horst, or basement high (Fig. 1a). The Helvetic nappes have been thrust above the Infrahelvetic units. In the studied region, the Helvetic nappe is termed the Wildhorn super-nappe that can be subdivided into the Diablerets, Mont Gond and Sublage nappes (Fig. 1c; Escher et al. (1993)). Due to the Rhone valley

associated with the Rhone-Simplon fault, the Helvetic nappes cannot be continuously traced back to their original position of deposition (Fig. 1c). The Ultrahelvetic units have been depositied on more distal regions than the Helvetic units (Fig. 1a). Today, the Ultrahelvetic units are found in front and between the Morcles and Wildhorn nappes (Fig. 1c).





During the Alpine continental collision, the Ultrahevetic units and the Penninic nappes, originating from more distal positions, have been thrust above the original deposition regions of the sediments forming today the Morcles and Wildhorn nappes (Fig. 1b) (e.g. Epard and Escher, 1996). These sediments were subsequently sheared off from their original position of deposition and were transported several tens of kilometers towards the foreland, i.e. top to the northwest transport direction (e.g. Epard and Escher, 1996; Ebert et al., 2007). The present day nappe structure represents a thick-skinned tectonic style because the crystalline basement of the Aiguilles-Rouges and Mont-Blanc massifs exhibits significant deformation (Fig. 1c).

The above tectonic scenario is supported by peak metamorphic temperatures of the Helvetic nappe system, which range between 250-385 °C (Kirschner et al., 1996, 1995; Ebert et al., 2007, 2008) increasing structurally downwards and towards the root zone. The maximal depth of burial of the Morcles nappe has most likely exceeded 10 km and was achieved between 29 Ma and 24 Ma (Fig. 1b) (Kirschner et al., 1996, 1995). In the studied section, the carbonate layers are strongly folded indicating significant internal deformation of the nappes (Fig. 1c). The Morcles fold nappe is characterized by strong parasitic folding in its frontal part and by a ca 20 km long, highly stretched inverse limb. The Wildhorn super-nappe also exhibits significant internal deformation, such as the isoclinal fold separating the Diablerets and Mont Gond nappes (Fig. 1c). These observations indicate that in the studied region ductile deformation was significant during formation of the nappes.

## 3 Methods

### 3.1 Mathematical model

Our mathematical model is based on the concept of continuum mechanics (e.g. Mase and Mase, 1970; Turcotte and Schubert, 2014). We assume slow, incompressible deformation under gravity so that inertial forces are negligible. Heat transfer by conduction, advection and production is considered. Heat transfer and deformation are coupled by shear heating, that is, dissipative deformation is converted into heat to conserve energy. The governing system of partial differential equations is solved numerically. The applied equations are described in detail in Schmalholz et al. (2019). The applied numerical algorithm is based on the finite-difference/marker-in-cell method (e.g. Gerya and Yuen, 2003). The diffusive terms in the force balance and heat transfer equations are discretized on an Eulerian staggered grid while advection and rotation terms are treated explicitly using a set of Lagrangian markers and a 4th order in space / 1st order in time Runge-Kutta scheme. The topography in the model is a material interface defined by a Lagrangian marker chain and this interface is displaced with the numerically calculated velocity field (Duretz et al., 2016). With ongoing deformation, this marker chain needs to be locally remeshed which is achieved by adding marker points in the deficient chain segments.

We consider a visco-elasto-plastic deformation behavior and assume a Maxwell viscoelastic model and Drucker-Prager yield criterion (see details in Schmalholz et al., 2019). In the applied creep flow laws, we add a constant pre-factor $f$ to vary the effective viscosities in order to test the impact of different effective viscosities on the model results. The effective viscosity is defined by the ratio of stress to (viscous) strain rate and has the form

$$\eta_{\text{eff}} = fFA^{-\frac{1}{n}}\dot{\epsilon}_{\text{II}}^{\frac{1}{n}-1}\exp\left(\frac{Q}{nRT}\right), \tag{1}$$





where the expression to the right of $f$ corresponds to the effective viscosity from standard creep flow laws determined by rock deformation experiments. Material parameters, such as effective viscosities, must be indepenent on the chosen coordinate system and, therefore, the dependence on strain rate is expressed by the quantity $\dot{\epsilon}_{\mathrm{II}}$, which is the square root of the second invariant of the (viscous) strain rate tensor. The equation for $\eta_{\mathrm{eff}}$ is given above because the magnitude and distribution of $\eta_{\mathrm{eff}}$

will be displayed for the performed simulations in the next section. All other parameters are explained and listed in Table 1.

## 3.2   Model configuration

The applied model configuration mimics a 200 km long section of the upper crustal region of a simplified passive margin (Fig. 2). We consider four model units with distinct mechanical properties, namely basement, cover, strong layer and weak unit. The basement unit represents the crystalline basement, the cover unit represents the Ultrahelvetic and Penninic nappes,

the strong layer represents the major carbonate layers (Malm and Urgonian) and the weak unit represents the shale-rich units. The initial geometry of the basement unit represents the crystalline upper crust of a passive continental margin with 15 km thickness, tapering down to 5 km thickness (Fig. 2). The Infrahelvetic basin is represented by an idealized half-graben that is 5 km deep and 25 km wide. The Infrahelvetic and the more distal (right model side) Helvetic basin are separated by an idealized horst structure. We cover the entire passive margin structure with sediments, to obtain a total (basement + sediments) model

thickness of 25 km. The model stratigraphy is consisting of three units (cover, strong layer and weak units) and each unit has homogenous material parameters. Both the half-graben and the basin are filled with weak units up to a depth of 13.5 km. On top of the weak units we place a 1.5 km thick strong layer. Our initial geometry represents the stage during the Alpine orogeny, when the proximal passive margin region, including the Infrahelvetic and Helvetic basins, is still relatively undeformed, but the Ultrahelvetic and Penninic units have been already thrust on top of it (Fig. 1b). We consider the overthrust units by adding

a 10 km thick, homogenous unit of cover sediments (without distinction between the Ultrahelvetic and Penninic units) on top of the model basement and basins (Fig. 2). Adding this 10 km thick unit is important to consider appropriately the ambient pressure and temperature, which control the brittle-plastic yield strength and the temperature dependent effective viscosities.

We apply boundary conditions that are similar to sandbox experiments of fold-and-thrust belts and orogenic wedges (Fig. 2). The left lateral model boundary moves to the right with a constant horizontal velocity of 1 cm.yr$^{-1}$, while the right lateral

boundary does not move horizontally. There are no shear stresses at the vertical model boundaries (i.e. free slip boundary conditions). The bottom boundary also moves with a horizontal velocity of 1 cm.yr$^{-1}$, but does not move vertically. This velocity boundary condition generates a velocity discontinuity at the bottom right corner of the model, which is typical for sandbox experiments and numerical simulations of accretionary wedges (e.g. Buiter et al., 2006). The top boundary is a free surface, using the algorithm of Duretz et al. (2016). We apply constant temperature boundary conditions of 10 °C at the top and

420 °C at the bottom of the model. There is no heat flux across the lateral model boundaries. We apply an initially equilibrated temperature field which results in a ca 16 °C.km$^{-1}$ initial geothermal gradient. Applied parameters are listed in Table 1.





## 4   Results

We present first the main results of a reference simulation, for the configuration described above, and then results of simulations in which some parameters are varied. All simulations show some common, general features: With increasing bulk shortening, the initially flat topography is increasing most around the right model boundary, representing a "back-stop" (Fig.

3). The models develop a wedge shape with a topography tilting towards the left model side. With progressive shortening, the increasing topography reaches the left model boundary and the topographic slope reduces, generating again a more horizontal topography. Also, initially the basement deformation occurs around the bottom right corner and progressively propagates towards the left (Fig. 3). With progressive shortening the models become generally thicker but the sedimentary units above the basement become relatively thicker than the underlying basement because the sediments are thrust above the basement. The

thickened sedimentary cover results in increasing basement temperatures, hence decreasing its teperature-dependent viscosity. The temperature increase of the top of the basement is visible by the vertical position of the 300 °C isotherm in figure (3). Such basement temperature increase and the related shift to a thick-skinned deformation was also reported by Bauville and Schmalholz (2015) in their numerical models of fold-and-thrust belts. Basement deformation results in the partial or total closure of the half-graben and associated extrusion of the basin fill. The specific model evolution, however, depends on the applied flow

laws and model stratigraphy, which will be discussed in the following in comparison with the reference simulation.

### 4.1   Reference model

We apply the configuration and parameters described in the previous section and displayed in figure 2 to generate a reference simulation for later comparison (Figs. 3 and 4). Initially, elastic stress builds up during a few hundred thousand years until the brittle-plastic yield stress and the steady-state ductile creep stress are reached. For the applied model configuration, the

brittle-ductile transition occurs at about 6-8 km depth. We quantify deviatoric stress magnitudes with the square root of the second invariant of the deviatoric stress tensor, $\tau_{\text{II}}$, and maximal deviatoric stresses reach ca 250 MPa at the brittle-ductile transition (Fig. 4). Maximal strain rates in the developing shear zones are between $10^{-13}$ and $10^{-12}$ s$^{-1}$, in broad agreement with strain rate estimates for natural shear zones (e.g. Pfiffner and Ramsay, 1982; Boutonnet et al., 2013; Fagereng and Biggs, 2018). The largest stresses occur around the brittle-ductile transition in the cover, whereas stresses in the basement and in the

strong layer are significantly smaller (Fig. 4).

The model shows several key phases of the formation of a nappe stack: (1) Detachment of a sedimentary unit of the right basin, mimicking the Helvetic basin, from their original substratum (Figs. 3a and b, and 4a and c). (2) Significant horizontal transport of ca 30 km with little internal deformation indicated by the relatively undeformed strong layer in the detached unit (Figs. 3b to d, and 4b to c). (3) Formation of a fold nappe due to ductile closure of the left half-graben, mimicking the Infrahelvetic basin, and associate extrusion of the sedimentary half-graben fill (Figs. 3c to e). (4) Stacking of the nappe

originating from the right basin above the fold nappe from the left half-graben (Figs. 3d and e).

During the initial stages of deformation, the strong layer of the right basin is gently folding, or buckling (Fig. 3a). Stress becomes concentrated around the contact of this strong layer and the basement horst (Fig. 5m) causing increased strain rates





in this region. With progressive deformation a localized shear zone, dominated by brittle-plastic deformation, develops across the strong layer, eventually detaching it from the basement (Fig. 5j to l). This shear zone develops within the strong layer so that a small piece of the strong layer remains attached to the basement (Fig. 5t). The detachment of the strong layer causes a significant stress drop in the strong layer and the basement (Fig. 5m to p). Once detached, the strong layer and parts of the

underlying weak unit passively move sub-horizontally over the horst initiating the horizontal nappe transport. Quantification of elastic strain rates shows that elastic deformation is active during the detachment process and that, hence, elastic stresses are not completely relaxed visco-plastically (Fig. 5e to h).

During the detachment, some parts of the weak cover, originally residing above the strong layer, are dragged below the detaching strong layer (Fig. 5). During the horizontal transport, the detached unit, consisting of the strong layer and some weak

units, is displaced above the cover material. Significant horizontal transport is facilitated because the underlying basement and the strong layer of the left half-graben are significantly more competent than the weak units at the base of the overthrusting nappe.

While the detached unit from the right basin is overthrusting the fill of the left half-graben, this fill is also sheared out of the half-graben due to (i) shear stresses generated by the overthrusting unit and (ii) closure of the half-graben due to ductile

deformation of basement units. During overthrusting, some parts of the cover units are incorporated between the overthrusting unit and the fill of the left half-graben. Finally, a nappe consisting of the fill from the right basin has been stacked above a fold nappe made of fill from the left half-graben. The entire process of nappe detachment, transport and stacking occurs during ca 8 Myr for the applied bulk shortening velocity of $1 \text{ cm.yr}^{-1}$. At the end of the simulation, the temperatures of the strong layer range between 250 °C at the topmost position and 350 °C in the root zone of the fold nappe. The final bulk shortening was ca

38 % after ca 8 Myr.

## 4.2 Impact of varying strength contrast

We performed three simulations with the same initial geometry as the reference simulation, but with modified pre-factors, $f$, in the applied flow laws. All simulations are terminated after a bulk shortening of ca 38 %, corresponding to the one of the reference simulation. In a first simulation, we used a smaller effective viscosity only for the basement ($f = 0.33$). Here,

the basement is weak enough to deform significantly from the beginning of shortening. Yield stresses are not reached at the contact of the basement horst with the strong layer (Fig. 6a). The strong layer does, hence, not detach from the basement and overthrusting does not take place. Instead, a several km large fold nappe develops in the strong layer of the right basin. Due to the highly distributed basement deformation, the half-graben closes only partially, resulting in a moderate buckling of the strong layer, but not in fold nappe formation. Also, a nappe stack does not form in the simulation.

In a second simulation, we used a stronger cover ($f = 0.5$ instead of $f = 0.1$). The effective viscosities of basement and cover are similar, hence, a mostly evenly distributed thick-skinned deformation is present from the beginning of shortening (Fig. 6c and d). A large scale fold develops above the horst, but the overturned limb made of the strong layer eventually detaches from the basement by necking. Although the overthrusting stage results in a significant horizontal displacement, this displacement is only half of the one in the reference simulation and not enough to form a nappe stack. Due to the stronger





shear drag from the top, the strong layer of the left half-graben is almost entirely sheared out from the half-graben. The strong layer of the left half-graben forms an overthrust nappe with significant horizontal displacement and with significant internal extension.

In a third simulation, we used weaker strong layers ($f = 0.33$). The development of the sediment units of the basin is largely similar to that in the reference model (Fig. 6). The only notable difference is that before the strong layer is detached from the basement, it forms a shear fold that is on the scale of a few km. The development of the units of the left half-graben is largely different compared to the reference simulation. Since the strong layer is weaker, the drag from the overriding units is sufficient to detach the strong layer from its left contact with the basement and displace it several tens of km to the left. During this displacement, the strong layer from the left half-graben is highly stretched and almost necking at its tail. Due to significant horizontal displacements, a nappe stack forms with two thrust sheets on the top of each other. However, the strong layer from the left half-graben is displaced considerably further towards the left than the strong layer from the right basin.

The final result of the three simulations, especially with respect to nappe detachment, transport and stacking, differs significantly from the result of the reference simulation, although the effective viscosities of individual model units have been modified by factors of only three to five (Fig. 6). The results indicate that the effective viscosity contrast between the model units has a first-order impact on the results.

At the onset of nappe formation, after ca 5% bulk shortening, the reference simulation and the three simulations with different $f$ factors exhibit different distributions and magnitudes of effective viscosity (Fig. 7). The maximal viscosity contrast in the reference simulation is up to five orders of magnitude inside the model domain, mainly between weak units in the basin and the uppermost cover (Fig. 7a). The effective viscosity at the top of the basement is in the order of $10^{24}$ Pa.s and the viscosity of the cover directly above the basement is at least one order of magnitude smaller. The strong layers have locally similar maximal effective viscosities than the top basement in the order of $10^{24}$ Pa.s. The effective viscosity contrast between strong layer and weak units in the right basin is ca three orders of magnitude (Fig. 7a). The above mentioned viscosity ratios between model units are required to generate the nappe detachment, transport and stacking in the reference simulation. In the three models with different $f$ factors in some model units, one of these viscosity ratios is different and, hence, the final result differs from the one of the reference simulation (Fig. 7).

### 4.3 Impact of multilayers

We also run simulations in which we replaced the single strong layer in the reference model with two thinner ones that are separated by weak units. We run three simulations with different initial thickness distributions of the two strong layers and alternating weak units. The initial thickness configuration is displayed on the right of the three panels in Fig. 8. The material parameters of every unit are the same as in the reference model. The basement deformation agrees with the one in the reference model. The deformation of the strong layers is different. Initially, the strong layers of the right basin form more intense, shorter wavelength (due to their smaller thickness) buckle folds, in agreement with the dominant wavelength theory (e.g. Biot, 1961; Schmalholz and Mancktelow, 2016).





In the simulations, where the upper strong layer rests directly below the cover (Fig. 8b, c), the top layer is being detached and transported in a similar fashion as in the reference model. The lower layer, on the other hand, forms a fold nappe first, with an extremely thinned inverse limb. Eventually this inverse limb develops boudinage, and once necking takes place it detaches from the basement. In the simulation, in which a weak unit is located between the upper strong layer and the cover (Fig. 8a),

both layers form folds and detach from the basement horst by necking in the inverse limb. After the detachment, the internal deformation of the strong layers is negligible.

The deformation of the units of the left half-graben is similar to the reference model, when weak units are located between the upper strong layer and the cover (Fig. 8a). The main difference to the reference simulation is that the weak unit located on top of the half-graben is sheared out, and both strong layers form a fold nappe with a more intensely stretched inverse limb. In

the models, in which the upper strong layer is in direct contact with the cover (Fig. 8b, c), the deformation of the strong layers in the half-graben is considerably different than it is in the reference model. Because the strong layer on the top of the left half-graben is much thinner, the drag from the overriding unit is sufficient to displace this layer considerably horizontally. The drag from the top shears the upper strong layer of the left half-graben above the basement to the left and it detaches the layer from the half-graben. As a result, buckle and shear folds form around the left tip of the layer (Fig. 8b). The upper strong layer

starts moving sub-horizontally without considerable internal deformation, and eventually forms a rootless nappe. The lower strong layer of the half-graben stays mostly in place, until the weak units are extruded from the half-graben due to its closure. Then, the lower strong layer forms a fold nappe, with a highly stretched inverse limb (Fig. 8b, c).

### 4.4 Impact of softening mechanisms

We also test the impact of two different softening mechanisms that can enhance strain localization (Fig. 9). The first mech-

anism is thermal softening by shear heating due to the conversion of mechanical work into heat and the resulting decrease of the temperature dependent viscosity (e.g. Yuen et al., 1978; Kaus and Podladchikov, 2006; Jaquet and Schmalholz, 2017; Kiss et al., 2019). Although this mechanism is activated in all simulations, for the applied 1 $\mathrm{cm.yr^{-1}}$ convergence velocity its impact on structure development is negligible. However, for faster bulk shortening, with 5 $\mathrm{cm.yr^{-1}}$ convergence velocity, thermal softening is sufficient to cause spontaneous shear zone formation (Kiss et al., 2019). For the simulation with 5 $\mathrm{cm.yr^{-1}}$

convergence velocity, all other parameters are identical to the ones in the reference simulation. There are two striking differences between the two simulations. First, in the high velocity simulation prominent ductile shear zones are formed in the cover that also promote the appearance of more localized brittle deformation zones (Fig. 9a). Second, heat production in the ductile shear zone raises the temperature of the units close to the "back-stop" on the right model side. Thus, the basement deformation in the right model domain is more intense, the left half-graben in the basement is not being closed and the sediment fill is not

being squeezed out (Fig. 9a).

The other considered softening mechanism is plastic strain softening. Such softening is frequently applied in numerical models of crustal deformation in order to enforce highly-localized brittle deformation by decreasing the friction angle as a function of accumulated plastic strain (e.g. Buiter et al., 2006). Such softening algorithm generates mesh-dependent results, but we apply it here for comparison, because such strain softening is applied in many numerical models of fold-and-thrust belts





(e.g. Buiter et al., 2006; Erdős et al., 2014; Ruh et al., 2014). We used two different parameter sets to model strain softening. In the first case (Fig. 9b), we start with a friction angle of 30° that we linearly decrease to 5° between an accumulated plastic strain of 0.5 and 1.5. Compared to the reference simulation, we observe strongly localized brittle deformation that is characterized by high angle ($\gg$ 0° from horizontal) and small displacement (< 10 km) overthrusts. This is the only simulation, where a strong

back-thrust forms over the right basin that also deforms the strong layer. In the second case (Fig. 9c), we start with a friction angle of 15° that we linearly decrease to 5° between accumulated plastic strain of 0.5 and 1.5. Such initially lower friction angle is often suggested to mimic fluid-pressure reduced effective friction angles (e.g. Erdős et al., 2014). In this simulation, the detachment mechanism of the right, strong layer from the horst is different, as the initial buckling and folding phase is entirely missing, and plastic yielding dominates from the beginning of deformation. Initially, the angle of thrusting is ca 35°

from the horizontal. Once a sufficient amount of weak units are sheared on top of the horst, the transport direction is sub-horizontal. Similarly to the other simulations with significant softening mechanisms, the basement around the half-graben is only deformed to a small degree and the half-graben is not closed.

## 5 Discussion

### 5.1 Numerical robustness

We investigated the impact of different numerical resolutions on the model results to test the robustness of these results. Such resolution test is important for the presented simulations, because weak material, such as cover and weak units, is entrapped along thin regions between stronger model units, such as strong layer and basement. Entrapment of weak material between strong material can cause mechanical decoupling if resolved numerically. We compare the reference model with an original resolution of 3001×1001 (width×height) numerical grid points (initially 66×25 m grid spacing) with two simulations having

identical configuration and parameters, but with smaller resolutions of 1501×501 (initially 133×50 m) and 751×251 (initially 267×100 m). The resulting structures after 38% of bulk shortening are essentially identical (Fig. 10). Similarly, the strain rate fields below the brittle-ductile transition are similar too. However, the strain rate distribution in the brittle part and around the brittle-ductile transition is resolution dependent (Fig. 10). This is typical for the applied non-associated plasticity scheme with the Drucker-Prager yield criterion, that is merely a stress limiter, inhibiting the stresses to exceed the failure limit. Thus the

exact geometry of the brittle-plastic shear bands is resolution dependent, but the effective load bearing capacity of the brittle layer converges with increasing resolution (Yamato et al., 2019, their appendix). Keeping in mind these limitations regarding the brittle-plastic deformation, the results in our main area of interest, that is the ductile nappe stacking, are essentially independent on the resolution within the studied range. Hence, our results are numerically robust concerning the detachment, transport and stacking of nappes under dominantly ductile deformation.





### 5.2 Comparison of the model results with the geological observations

There are several features of the Helvetic Nappe System that we could successfully reproduce in our thermo-mechanical model. Similarly to Bauville and Schmalholz (2015), a structure resembling a fold nappe has been formed by the extrusion of the sedimentary fill from a half-graben. During formation of this fold nappe, the half-graben has been closed and the sediments

squeezed between the two basements resemble the structure of the Chamonix zone located between the Aiguilles-Rouges and Mont-Blanc massifs (Fig. 1c). Hence, our model generated the first-order structural features of the Infrahelvetic complex in W-Switzerland, namely a recumbent fold nappe with a root located between two deformed basement massifs. Additionally, our model reproduced the detachment and sub-horizontal transport of sedimentray units from a model passive margin structure. The thrust nappe, which originates from the rigth basin in our model, resembles the Wildhorn super-nappe. The horizontal

transport of this thrust nappe is in the order of 30 km in the model. Furthermore, in the model this thrust sheet is stacked above the fold nappe and the final model structure resembles a thrust nappe that is stacked above a fold nappe, as observed in the Helvetic nappe system. Moreover, there is a considerable amount of cover units entrapped between the fold nappe and the thrust sheet. The entrapped lower region of the cover unit resembles the Ultrahelvetic units so that our model can explain how these Ultrahelvetic units have been entrapped between the Morcles fold nappe and the Wildhorn super-nappe (Fig. 1c).

At the end of the simulated formation of the nappe system, the maximum temperature in the nappe system ranges between 250 °C and 350 °C, which is in agreement with the metamorphic peak temperatures of the Helvetic nappe system reported by Kirschner et al. (1996) and Ebert et al. (2007). In the simulations, the nappe stack is formed within ca 8 Myr, which is also in broad agreement with the estimated time span of main formation of the Morcles fold nappe from ca 28 to 17 Ma (Kirschner et al., 1995). The simulations with two thin strong layers, separated by weak units, can explain the significant parasitic, or

second order, folding of the two main carbonate units (Quinten and Urgonian limestone formations) which is observed in the Wildhorn super-nappe.

Some features of the Helvetic Nappe System are not reproduced by the simulations. In the frontal part of the fold nappe, originating from the left half-graben, the front is first thrust out of the half-graben and the overturned limb develops subsequently. This deformation generates a "nose-like" structure in the frontal part of the fold nappe, which is not observed. Also, in

all simulations the fold nappe has only a minor second order folding, in contrast with the prominent parasitic folds of the Morcles nappe. Reproducing these natural, second-order parasitic folds with their correct scale with respect to the first-order fold nappe would require an even higher numerical resolution. In the numerical models, we also likely overestimated the amount of shale-rich sediments in the right basin, mimicking the Helvetic basin, as the total volume of the Wildhorn super-nappe south of the Morcles nappes is much thinner than in the simulations. There was also likely a significant amount of vertical flattening,

and presumably pressure solution related volume decrease, after the main phase of nappe formation and during the exhumation of the nappe system, which is not modelled in our simulations. Moreover, several basement shear zones have been mapped in the Aiguilles-Rouges and in the Mont-Blanc massifs, which are not present in the simulations. This is likely because (i) the straight bottom boundary of the model may prohibit any significant vertical displacement of the basement units and hence inhibit significant shear zone formation, (ii) the model basement is mechanically homogeneous and there are no heterogeneities





that can trigger shear zone localization and (iii) the amount of brittle-plastic deformation is underestimated in the basement. We considered a horizontal model base while during natural nappe formation the overall basement-cover interface was likely dipping, or tilting, in the direction of subduction (i.e. direction of basal velocity), so that a model bottom inclinded towards the subduction direction would be more realistic. The deformation at our model bottom is viscous and the surface slope for

evolving crustal wedges with a viscous base depends on the viscous shear stress at the base, whereby larger shear stresses are related to higher surface slopes (e.g. Ruh et al., 2012). Keeping the basal viscous shear stress the same, a tilting of the model base towards the subduction direction would reduce the surface slope. Therefore, in our models the surface slopes towards the foreland (left) region represent high end-member surface slopes so that effects of gravity-related forces directed towards the foreland region are on the higher end.

Finally, the applied "numerical sandbox" model configuration and velocity boundary conditions constrain the deformation in the model domain. During the large-scale dynamics of Alpine orogenic wedge formation, the straight bottom and right model boundaries do not exist. Processes such as laterally-varying vertical isostatic adjustment, flexure due to subduction and back-thrusting, or back-folding, generate geodynamic conditions for the formation of the Helvetic nappe system which are clearly more dynamic and complex than implied by the considered model configuration. Lithospheric scale numerical models can

self-consistently model the generation of orogenic wedges and major crustal shear zones, including effects of isostasy, flexure and back-folding (e.g. Erdős et al., 2014; Jaquet et al., 2018; Jourdon et al., 2019; Erdős et al., 2019). With higher numerical resolution such lithosphere models may eventually be able to resolve the upper crustal deformation with a resolution as applied in our model.

### 5.3    Tectonic inheritance, mechanical heterogeneities and potential softening mechanisms

Geological reconstructions of the Helvetic nappe system showed the correlation of the nappes with their original positions along the pre-Alpine European passive margin, which was characterized by half-grabens and horsts (e.g. Epard, 1990; Boutoux et al., 2014). In agreement with previous modelling studies (e.g. Beaumont et al., 2000; Wissing and Pfiffner, 2003; Bellahsen et al., 2012; Lafosse et al., 2016; Bauville and Schmalholz, 2017), our results suggest that tectonic inheritance in the crust in the form of half-grabens and horsts has a strong impact on the development of fold and thrust nappes during crustal deformation.

Our results indicate that two features of the tectonic inheritance are important, namely the geometry and the magnitude of mechanical heterogeneities. The geometry of half-grabens and horsts controls the location of nappe initiation (Bauville and Schmalholz, 2017). The basement and sediments must, of course, have different mechanical strength, otherwise the geometry of the basement would be unimportant. Our results suggest that tectonic inheritance is necessary to model the evolution of the Helvetic nappe system, but not sufficient. The results show that specific strength, or effective viscosity, contrast between

basement and sediments and within the sediments are required to model nappe structures resembling those of the Helvetic nappe system (Figs. 6 and 7). The reference simulation exhibits a viscosity contrast between weak units and strong layer and basement in the order of four orders of magnitude (Fig. 7a). Although the effective viscosity in the basement and strong layer is in the order of $10^{24}$ Pa.s, the stresses in the basement and strong layer are far below the brittle-plastic yield stress and typically smaller than 100 MPa (Fig. 4). If the effective viscosity contrast between strong layer and basement is not large enough,





then the sediments of the right basin are not detached in the manner of a thrust sheet from their original position (Fig. 6a). One possibility to enforce detachment also for smaller viscosity contrasts is the application of plastic strain softening and/or initially reduced friction angles (Fig. 9b and c). Application of strain softening favors the formation of thrust sheets in the models, but prohibits the formation of fold nappes (Fig. 9b and c). The importance of tectonic inheritance and the pre-Alpine configuration

on the nappe formation during Alpine orogeny underlines the importance of geological field work and associated geological reconstructions, because only such field based reconstructions can provide estimates for the pre-Alpine configurations. Our results are consistent with those of Duretz et al. (2011) which showed that inherited mechanical heterogeneities, promoting large lateral strength contrast, are essential to trigger exhumation of lower crustal granulites as observed in the Bohemian Massif. Generally, our results are consistent with a variety of studies, which show the importance of structural inversion of

extensional systems during compressional deformation and are based on geological field observation, analogue deformation experiments and numerical models (e.g. Gillcrist et al., 1987; Buiter and Adrian Pfiffner, 2003; Buiter et al., 2009; Bellahsen et al., 2012; Bonini et al., 2012; Lafosse et al., 2016; Granado and Ruh, 2019).

For the applied model configuration, a significant localization due to thermal softening does not occur for a convergence velocity of 1 $cm.yr^{-1}$, but it does for 5 $cm.yr^{-1}$. Average convergence velocities during the Alpine orogeny are typically

estimated to be in the order of 1 $cm.yr^{-1}$ (Schmid et al., 1996). However, some short periods with higher convergence velocities cannot be excluded. So if there were short periods during the formation of the Helvetic nappe system with convergence velocities larger than ca 5 $cm.yr^{-1}$, then thermal softening might have been important.

There is field evidence for grain size reduction associated with mylonitic shear zones at the base of nappes in the Helvetic nappe system (e.g. Ebert et al., 2007, 2008). We did not consider the microscale grain size reduction in our models for several

reasons: First, the major mylonitic shear zones with significant grain size reduction have a thickness in the order of 10 m. Although we use high resolution models we have a numerical grid size of ca $66 \times 25$ m, hence, this resolution is still not large enough to resolve the internal deformation within shear zones having thickness of 10 m. Second, recent numerical simulations including grain size reduction and combined diffusion and dislocation creep flow laws suggest that grain size reduction does not have a dramatic impact on strain localization (Schmalholz and Duretz, 2017), which is in agreement with theoretical

results of Montési and Zuber (2002). The reason is that a piezometer-type stress to grain size relation, when subsituted into a grain-size-sensitive diffusion creep flow law, generates a power-law type flow law with stress exponents similar to the one of the corresponding dislocation creep flow law (e.g. Montési and Zuber, 2002). However, other studies argue that microscale processes such as coupled grain evolution and damage mechanisms can generate significant strain localization and that these mechanisms have been responsible for generating subduction and plate tectonics (e.g. Bercovici and Ricard, 2014). Therefore,

future simulations should consider such coupled microscale processes in order to quantify their importance on the first order tectonic nappe detachment, overthrusting and stacking.





# 6  Conclusions

The presented 2D thermo-mechanical simulations of shortening of the upper crustal region of a passive margin consider initial mechanical heterogeneities and can explain key aspects of tectonic nappe formation. Two types of heterogeneities are considered: (1) a lateral heterogeneity due to the basement-cover interface that is characterized by half-grabens and horsts, and (2)

a vertical heterogeneity due to alternating sedimentary layers with different mechanical strength. The simulations can model the detachment and sub-horizontal transport of a nappe and the stacking of two nappes. Detachment of sedimentary units occurs due to stress concentrations in strong sediment layers around the basement-sediment contact, which result in a localized brittle-plastic shear band that detaches the strong layer from the basement. Horizontal transport is controlled by basement-cover geometry and occurs by moving relatively stronger sediment units above thinner and weaker units. The detached and

horizontally transported units resemble a thrust nappe. Structures resembling fold nappes form by the ductile closure of a half-graben and the associated extrusion of the sedimentary half-graben fill; in agreement with previous modelling studies. In the simulations, the thrust nappe initiates before the fold nappe, because it is located closer to the "back-stop" of the applied model configuration. The thrust nappe exhibits a larger horizontal displacement than the fold nappe so that the thrust nappe can be stacked above the fold nappe.

The considered lateral variation of the basement-cover interface and associated mechanical heterogeneity is necessary to model nappe formation and stacking, but it is not sufficient. Additionally, a specific effective viscosity contrast between basement and strong and weak sediments is required. For our model configuration, an effective viscosity contrast of approximately four orders of magnitude between weak sediments and strong sediments and basement is required. Simulations with smaller viscosity contrasts did not generate the detachment and transport of thrust nappes. Nappe detachment and transport are mod-

eled with standard creep flow laws and a Drucker-Prager yield criterion, without the application of strain softening algorithms. Considering several strong layers in the models can explain the second-order internal folding observed within some thrust nappes.

Based on the first-order agreement between our model results and natural data, we propose a macroscale "nappe theory" for the Helvetic nappe system of Western Switzerland. Our "nappe theory" can be self-consistently calculated and reproduced

with a well-established continuum mechanics thermo-mechanical theory and with standard creep flow laws and a Drucker-Prager yield criterion, which are based on rock deformation experiments. We propose that the pre-Alpine configuration of the European passive margin was characterized by important mechanical heterogeneities resulting from (i) a basement-cover contact with half-grabens and horst, and (ii) the alternation of mechanically strong and weak sediment units. During the Alpine continental collision, the passive margin is shortened and sheared due to external compressive stresses. During margin

deformation, mechanical heterogeneities control the detachment, transport and stacking of nappes. The sedimentary units of the Wildhorn super-nappe are detached due to the existence of horsts, or basement highs, causing stress concentrations and brittle-plastic shear bands. The Wildhorn super-nappe was transported mainly above weak shale-rich units and weak Ultrahelvetic units, which have been entrapped from above to below the Wildhorn super-nappe. Formation of the Morcles fold nappe occured





by dominantly ductile closing of a half-graben, bounded by basement massifs that form now the Aiguilles-Rouges and Mont-Blanc massifs, and the associated squeezing-out of sediments from this half-graben.

*Code availability.* Available upon request from T.D.

*Data availability.* Available upon request from D.K.

5 *Author contributions.* D.K. carried out the numerical simulations, visualized and interpreted the results and prepared the first draft, T.D. developed the numerical algorithm, implemented the model configuration and contributed to the visualization, interpretation and presentation of the results. S.M.S. designed the original research project and contributed to the visualization, interpretation and presentation of the results.

*Competing interests.* No competing interests are present.

*Acknowledgements.* This work was supported by SNF grant No. 200020-149380 and the University of Lausanne.





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



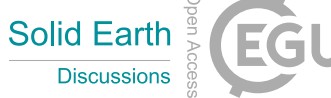

**Figure 1.** a) Simplified geological reconstruction of the Mesozoic, pre-Alpine European passive margin. b) Simplified geological reconstruction of the Alpine orogenic wedge, after emplacement of the Penninic and Ultrahelvetic units and before the Helvetic nappe stacking. The black rectangle represents the model domain of the numerical simulations. c) Simplified geological reconstruction of the present day structure of the Helvetic Nappe System.





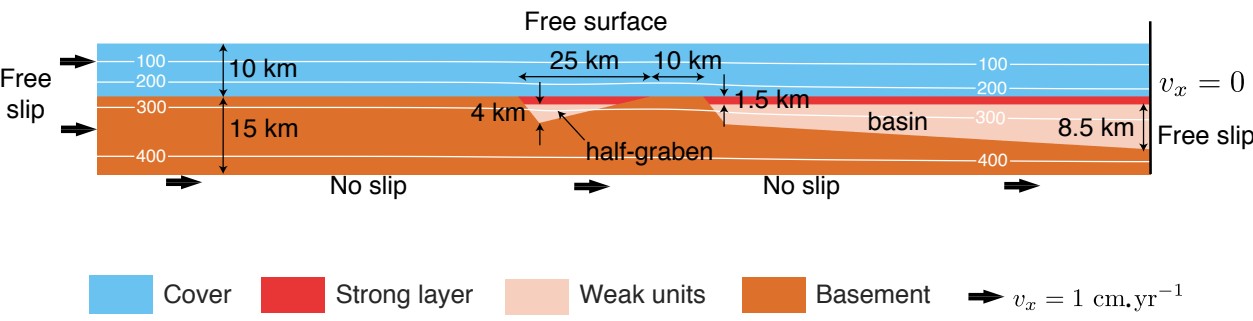

**Figure 2.** Reference model configuration. The white contours show isotherms and the labels are the corresponding temperatures in °C.





**Figure 3.** Structural and thermal evolution of the reference model for different times, $t$, and bulk shortening, $\gamma_b$. The white contours show isotherms and the labels are the corresponding temperatures in °C.





**Figure 4.** Evolution of the strain rate (left column) and deviatoric stress (right column) fields of the reference model for different times, $t$, and bulk shortening, $\gamma_b$. Strain rate and deviatoric stress are quantified with the square root of the second invariant of the strain rate, $\dot{\epsilon}_{\mathrm{II}}$, and deviatoric stress, $\tau_{\mathrm{II}}$, tensor, respectively. Magnitudes of $\dot{\epsilon}_{\mathrm{II}}$ and $\tau_{\mathrm{II}}$ are displayed with logarithmic colorscale. Colormaps are from Crameri (2018).





**Figure 5.** Enlargement of different stages of the detachment of the strong layer from the basement horst for different times, $t$, and bulk shortening, $\gamma_b$ (see figure 4 for entire model domain). Colorplots of viscous strain rates (a-d), elastic strain rates (e-h), plastic strain rates (i-l), deviatoric stresses (m-p) and effective viscosities (q-t) are displayed. For all strain rate and stress tensor quantities we display their corresponding square root of the second invariants.







**Figure 6.** The final geometry, temperature and viscosity fields of three simulations with different $f$ factor for certain model units (see text).





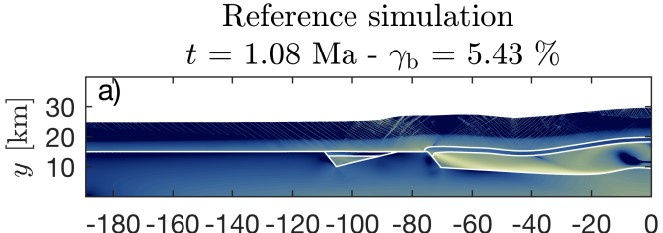

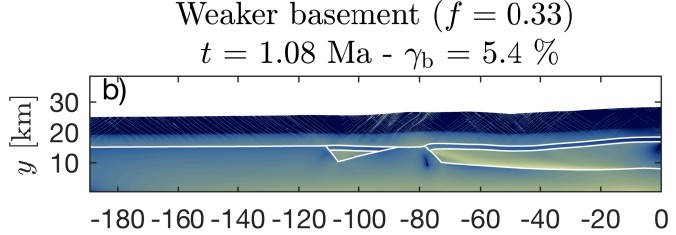

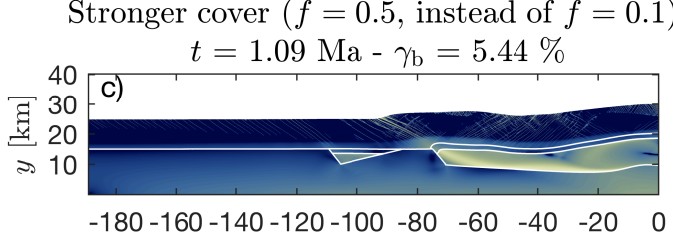

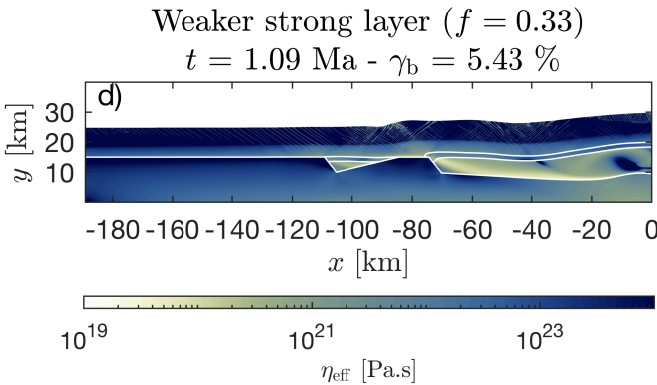

**Figure 7.** Effective viscosity for four simulations with different $f$ factor for certain model units after a bulk shortening of ca 5.4%. Panel a) displays the reference simulation and panels b) to d) displays the three simulations shown in figure 6.

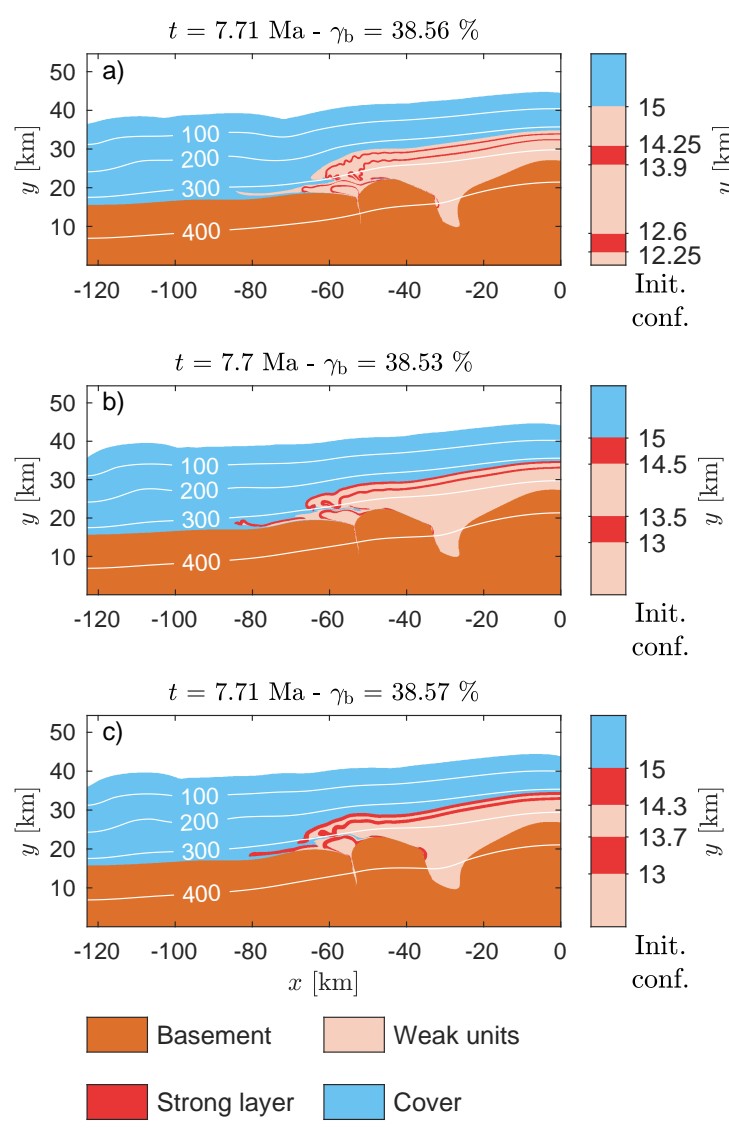

**Figure 8.** The final geometry of three simulations with two strong layers with the isotherms of the corresponding temperature field. The initial model stratigraphy around the upper region of the half-graben and basin is displayed on the right of each panel. The model stratigraphy is laterally homogenous, so the overall initial configuration is similar to that in Figure (2).





**Figure 9.** The geometry and the strain-rate field of three simulations after ca 30% bulk shortening, with various softening mechanisms. Panels a) and b) show results of a simulation with a convergence rate of 5 $cm.yr^{-1}$, in which thermal softening has a significant impact. Panels c) and d) show results of a simulation with strain softening that reduces friction angle from the initial 30 degrees to 5 degrees. Panels e) and f) show results of a simulation with strain softening that reduces friction angle from the initial 15 degrees to 5 degrees.





**Figure 10.** The geometry and the strain-rate field of three simulations after ca 38% bulk shortening for different numerical resolutions.



**Table 1.** The list of the reference model parameters, where $f$ is a custom pre-factor, $A$ is the pre-exponential factor, $n$ is the power-law exponent, $Q$ is the activation energy, $\lambda$ is the thermal conductivity, $\rho_{\mathrm{ref}}$ is the density at reference pressure ($P_{\mathrm{ref}} = 0$ Pa) and temperature ($T_{\mathrm{ref}} = 0$ °C), $Q_r$ is the radioactive heat production, $C$ is the cohesion and $\phi$ is the friction angle. Some parameters have constant values: $C_p = 1050$ J.K$^{-1}$ is the heat capacity, $G = 10^{10}$ Pa is the shear modulus, $\alpha = 3 \times 10^5$ K$^{-1}$ is the thermal expansion coefficient, $\beta = 10^{-11}$ Pa$^{-1}$ is the compressibility and $F = 2^{(1-n)/n} 3^{-(n+1)/(2n)}$ is a geometry factor (needed to convert flow law parameters from axial compression experiments into an invariant form). The creep flow law parameters ($A$, $n$ and $Q$) are: [1]Westerly granite (Hansen et al., 1983), [2]calcite (Schmid et al., 1977) and [3]mica (Kronenberg et al., 1990).

| Lithology | $f$ | $A$ [Pa$^{-n}$s$^{-1}$] | n | $Q$ [J.mol$^{-1}$] | $\lambda$ [W.m$^{-1}$K$^{-1}$] | $\rho_{\mathrm{ref}}$ [kg.m$^{-3}$] | $Q_{\mathrm{r}}$ [W.m$^{-3}$] | $C$ [Pa] | $\phi$ [°] |
|---|---|---|---|---|---|---|---|---|---|
| Basement[1] | 1.0 | $3.16 \times 10^{-26}$ | 3.3 | $1.87 \times 10^5$ | 3.0 | 2800 | $2.5 \times 10^{-6}$ | $10^7$ | 30 |
| Cover[2] | 0.1 | $1.58 \times 10^{-25}$ | 4.2 | $4.45 \times 10^5$ | 2.5 | 2700 | $5 \times 10^{-7}$ | $10^7$ | 30 |
| Strong layer[2] | 1.0 | $1.58 \times 10^{-25}$ | 4.2 | $4.45 \times 10^5$ | 2.5 | 2750 | $5 \times 10^{-7}$ | $10^7$ | 30 |
| Weak units[3] | 1.0 | $1.00 \times 10^{-138}$ | 18.0 | $5.10 \times 10^4$ | 2.0 | 2700 | $1 \times 10^{-6}$ | $10^6$ | 5 |