# Peer review of "Tectonic inheritance controls nappe detachment, transport and stacking in the Helvetic Nappe System, Switzerland: insights from thermo-mechanical simulations"

_Solid Earth, 2019_

## Referee Comment (RC1) · Luca Dal Zilio (Referee) · 14 Oct 2019

"Towards a nappe theory: Thermo-mechanical simulations of nappe detachment, transport and stacking in the Helvetic Nappe System, Switzerland" by Kiss and colleagues is an interesting paper that investigate the thermo-mechanical processes of nappe formation. Overall, the paper is quite short, but well-written, pretty balanced, and the illustrations are to the point. It provides a modern and clear perspective on the topic. As soon as the authors consider the comments below, I will be happy to recommend this work for publication in EGU Solid Earth. I think with some improvements this review paper will be ready to have a big impact and long shelf-life.

[Figure]

GENERAL COMMENTS:

- I found the title "Towards a nappe theory" and the first part of the introduction a bit far from the aim of this study. This study is definitely a step towards a better theory of tectonic nappes; however, it is focused on a specific case (Helvetic Nappe System) and the model setup is also made for it. Based on my comment, I suggest to remove "Towards a nappe theory" from the title and rephrase the first part of the introduction (see my next comment).

- The introduction is very detailed. The authors provide a very broad overview. I would recommend to make it shorter. Also, the authors go back and forth between the general knowledge on the topic and what is addressed in the study. I suggest to separate this parts and improve the transition between the two; for examples, I would add some lines to highlight how numerical simulations can help to overcome the uncertainties from e.g., geological interpretation and/or typical limitation of analogue models.

SPECIFIC COMMENTS:

Page 1:

**5: "of a thrust nappe and stacking of this thrust nappe" - remove "of this thrust nappe"?**

**10: "and the resulting brittle-plastic shear band formation" - shear band (bands?) cutting through the cover layer?**

**10: "weak sediments" - décollement?**

Page 2: #5 ", for example, a basic definition" - ; for example. . .

Page 5:

**15 "We assume slow, incompressible deformation" - please be more specific with the term "slow". Maybe long-term tectonic deformation?**

**25 "With ongoing deformation, this marker chain needs to be locally remeshed which**

is achieved by adding marker points in the deficient chain segments." - The term remesh is odd, as it refers to the "Lagrangian" markers. Please specify whether this criterion assumes a minimum number of markers per cell. If so, please clarify how these markers are added and how the physical properties are interpolated from the nodes.

Page 6:

**20 "ambient pressure and temperature"?**

**25 "The top boundary is a free surface, using the algorithm of Duretz et al. (2016)". I recommend to spend a few more lines to specify how this algorithm works and that this is not "the usual" pseudo free-surface used in many geodynamic models.**

**25 I suppose the velocity discontinuity at the bottom right corner introduces a stress singularity - how do you treat this issue in the boundary conditions?**

Page 7:

**20 "deviatoric stresses reach ca 250 MPa". This values seems pretty high. In section 4.4 the authors discuss the effects of softening mechanisms - e.g., lower effective friction to mimic the presence of pore fluid-pressure. I was wondering what is level of deviatoric stresses when the model is under hydrostatic conditions.**

Figure 1: could you please add a small inset to locate the region of the cross-section?

I hope my comments contribute the authors to improve the manuscript.

Luca Dal Zilio

---

## Referee Comment (RC2) · Patrice Rey (Referee) · 25 Nov 2019

The paper presents a set of high-resolution thermo-mechanical simulations aiming towards a "nappe theory". The simulations focus strongly on reproducing as many features and attributes documented in the Helvetic nappe system, which guides the choice of input parameters, geometry, and boundary conditions. From a reference simulation, a set of key parameters are varied to test their influence on the simulation outcome. • First, the viscosity of each material is tested in turn by (i) dropping the viscosity of the basement, (ii) increasing the viscosity of the cover sequence, and (iii) increasing the viscosity of the stronger syn-rift unit capping the rift basins. • Then the stronger

syn-rift unit is replaced by a 4- or 5-layer system involving 2 stronger layers and 2 or 3 weaker units. Three simulations test various thicknesses and configurations. • Two strain weakening mechanisms (i.e. shear heating and accumulated plastic strain) are tested at various extensional velocities (1 cm/yr and 5 cm/yr). The simulation outcomes are then compared to the Helvetic nappe system.

The paper will be of great interest to geologists interested in nappe tectonics and in particular those interested in the Helvetic nappe system. The paper is well organized, relatively easy to follow, and the figures serve their purpose reasonably well.

The study is an attempt to learn about nappe tectonics from reproducing via numerical simulations the well-documented Helvetic nappe system. However, it remains to be seen whether an all-encompassing "nappe theory" can be extracted from such an approach, for two reasons: • I would first question in the present context the use of the word "theory". In natural science, a theory is a very robust model established over decades of data collection and analysis and explaining a very large range of unrelated observations. Plate tectonics and biological evolution are two theories. For this reason, I think that the concept of "nappe theory" could safely be replaced by the concept of "nappe model". • In addition, I think it is pretty safe to state that there is probably more than one way for nappes to develop. Hence, the proposed model is only strictly relevant to the Helvetic nappe system that develops as the result of the inversion of an extended continental margin, and the extrusion of its syn-rift sedimentary infilling. Hence, I think that modifying slightly the title and introduction, to bring a stronger focus on the "Helvetic style" of nappe tectonics, would be beneficial to the paper.

Perhaps the main missing ingredient in the numerical experiments presented here is isostasy and the absence of flexure despite up to 10 km of topography due to crust thickening and nappe stacking. I acknowledge that this issue is touch upon in section 5.2, but it is important to stress that the outcome of this set of simulation will change should the basement be allowed to subside under the weight of the nappe stack.

[Figure]

The paper would also benefit from being leaner. I found at places the paper to be unnecessarily wordy, and the description on the simulation lengthy and tedious to read. Rather than describing the evolution of each experiment in great detail (perhaps you can point toward movies or animations instead), it would be best to highlight key differences. The conclusion needs to be rewritten and shortened. A conclusion goes beyond merely repeating what was said before.

The supplementary section needs some editing, there are too many spelling mistakes.

Finally, either a code is made freely available, or it is not. Having to ask permission to the author to access the code is, in my view, not sufficient. Codes which are accessible are available online (e.g. underworldcode.org). Chances are that in ten years Underworld will still be available like it was ten years ago.

Kind regards,

Patrice Rey

Please also note the supplement to this comment:
https://www.solid-earth-discuss.net/se-2019-130/se-2019-130-RC2-supplement.pdf

**Supplement:**

[revised manuscript text omitted]

**Reference simulation**
**$t = 1.08$ Ma - $\gamma_{\mathrm{b}} = 5.43$ %**

[Figure]

**Weaker basement $(f = 0.33)$**
**$t = 1.08$ Ma - $\gamma_{\mathrm{b}} = 5.4$ %**

[Figure]

**Stronger cover $(f = 0.5,$ instead of $f = 0.1)$**
**$t = 1.09$ Ma - $\gamma_{\mathrm{b}} = 5.44$ %**

[Figure]

**Weaker strong layer $(f = 0.33)$**
**$t = 1.09$ Ma - $\gamma_{\mathrm{b}} = 5.43$ %**

[Figure]

**Figure 7.** Effective viscosity for four simulations with different $f$ factor for certain model units after a bulk shortening of ca 5.4%. Panel a) displays the reference simulation and panels b) to d) displays the three simulations shown in figure 6.

[Figure]

**Figure 8.** The final geometry of three simulations with two strong layers with the isotherms of the corresponding temperature field. The initial model stratigraphy around the upper region of the half-graben and basin is displayed on the right of each panel. The model stratigraphy is laterally homogenous, so the overall initial configuration is similar to that in Figure (2).

[Figure]

**Figure 9.** The geometry and the strain-rate field of three simulations after ca 30% bulk shortening, with various softening mechanisms. Panels a) and b) show results of a simulation with a convergence rate of 5 cm.yr$^{-1}$, in which thermal softening has a significant impact. Panels c) and d) show results of a simulation with strain softening that reduces friction angle from the initial 30 degrees to 5 degrees. Panels e) and f) show results of a simulation with strain softening that reduces friction angle from the initial 15 degrees to 5 degrees.

[Figure]

**Figure 10.** The geometry and the strain-rate field of three simulations after ca 38% bulk shortening for different numerical resolutions.

**Table 1.** The list of the reference model parameters, where $f$ is a custom pre-factor, $A$ is the pre-exponential factor, $n$ is the power-law exponent, $Q$ is the activation energy, $\lambda$ is the thermal conductivity, $\rho_{\mathrm{ref}}$ is the density at reference pressure ($P_{\mathrm{ref}} = 0$ Pa) and temperature ($T_{\mathrm{ref}} = 0$ °C), $Q_r$ is the radioactive heat production, $C$ is the cohesion and $\phi$ is the friction angle. Some parameters have constant values: $C_p = 1050$ J.K$^{-1}$ is the heat capacity, $G = 10^{10}$ Pa is the shear modulus, $\alpha = 3 \times 10^5$ K$^{-1}$ is the thermal expansion coefficient, $\beta = 10^{-11}$ Pa$^{-1}$ is the compressibility and $F = 2^{(1-n)/n}3^{-(n+1)/(2n)}$ is a geometry factor (needed to convert flow law parameters from axial compression experiments into an invariant form). The creep flow law parameters ($A$, $n$ and $Q$) are: [1]Westerly granite (Hansen et al., 1983), [2]calcite (Schmid et al., 1977) and [3]mica (Kronenberg et al., 1990).

| Lithology | $f$ | $A$ [Pa$^{-n}$s$^{-1}$] | n | $Q$ [J.mol$^{-1}$] | $\lambda$ [W.m$^{-1}$K$^{-1}$] | $\rho_{\mathrm{ref}}$ [kg.m$^{-3}$] | $Q_{\mathrm{r}}$ [W.m$^{-3}$] | $C$ [Pa] | $\phi$ [°] |
|---|---|---|---|---|---|---|---|---|---|
| Basement[1] | 1.0 | $3.16 \times 10^{-26}$ | 3.3 | $1.87 \times 10^5$ | 3.0 | 2800 | $2.5 \times 10^{-6}$ | $10^7$ | 30 |
| Cover[2] | 0.1 | $1.58 \times 10^{-25}$ | 4.2 | $4.45 \times 10^5$ | 2.5 | 2700 | $5 \times 10^{-7}$ | $10^7$ | 30 |
| Strong layer[2] | 1.0 | $1.58 \times 10^{-25}$ | 4.2 | $4.45 \times 10^5$ | 2.5 | 2750 | $5 \times 10^{-7}$ | $10^7$ | 30 |
| Weak units[3] | 1.0 | $1.00 \times 10^{-138}$ | 18.0 | $5.10 \times 10^4$ | 2.0 | 2700 | $1 \times 10^{-6}$ | $10^6$ | 5 |

---

## Author Comment (AC1) · 10 Jan 2020

"Towards a nappe theory: Thermo-mechanical simulations of nappe detachment, transport and stacking in the Helvetic Nappe System, Switzerland" by Kiss and col- leagues is an interesting paper that investigate the thermo-mechanical processes of nappe formation. Overall, the paper is quite short, but well-written, pretty balanced, and the illustrations are to the point. It provides a modern and clear perspective on the topic. As soon as the authors consider the comments below, I will be happy to recommend this work for publication in EGU Solid Earth. I think with some improvements this review paper will be ready to have a big impact and long shelf-life.

GENERAL COMMENTS:

- I found the title "Towards a nappe theory" and the first part of the introduction a bit far from the aim of this study. This study is definitely a step towards a better theory of tectonic nappes; however, it is focused on a specific case (Helvetic Nappe System) and the model setup is also made for it. Based on my comment, I suggest to remove "Towards a nappe theory" from the title and rephrase the first part of the introduction (see my next comment).

We agree, therefore we modified the title accordingly.

- The introduction is very detailed. The authors provide a very broad overview. I would recommend to make it shorter. Also, the authors go back and forth between the general knowledge on the topic and what is addressed in the study. I suggest to separate this parts and improve the transition between the two; for examples, I would add some lines to highlight how numerical simulations can help to overcome the uncertainties from e.g., geological interpretation and/or typical limitation of analogue models.

We have shortened the introduction.

SPECIFIC COMMENTS:

Page 1:

**5: "of a thrust nappe and stacking of this thrust nappe" - remove "of this thrust nappe"?**

We reformulated it.

**10: "and the resulting brittle-plastic shear band formation" - shear band (bands?) cutting through the cover layer?**

We reformulated the sentence to make it clearer.

**10: "weak sediments" - décollement?**

We extended the sentence to make it clearer.

Page 2: #5 ", for example, a basic definition" - ; for example. . .

We modified the sentence accordingly.

Page 5:

**15 "We assume slow, incompressible deformation" - please be more specific with the term "slow". Maybe long-term tectonic deformation?**

We added the information that slow means here: no inertial forces.

**25 "With ongoing deformation, this marker chain needs to be locally remeshed which is achieved by adding marker points in the deficient chain segments." - The term remesh is odd, as it refers to the "Lagrangian" markers. Please specify whether this criterion assumes a minimum number of markers per cell. If so, please clarify how these markers are added and how the physical properties are interpolated from the nodes.**

We reformulated the sentence. The marker chain is actually a contour line defined by marker points, and these marker points are different to the markers that carry information on material properties.

Page 6:

**20 "ambient pressure and temperature"?**

We reformulated the sentence to make it clearer.

**25 "The top boundary is a free surface, using the algorithm of Duretz et al. (2016)". I recommend to spend a few more lines to specify how this algorithm works and that this is not "the usual" pseudo free-surface used in many geodynamic models.**

The algorithm is explained in a bit more detail in the mathematical model section. p.5. #25

**25 I suppose the velocity discontinuity at the bottom right corner introduces a stress singularity - how do you treat this issue in the boundary conditions?**

The velocity around the discontinuity is linearly decreased to zero within a small distance to minimize and smooth the effect of the discontinuity. The presented stress fields show that the stress perturbation around the discontinuity are minor.

Page 7:

**20 "deviatoric stresses reach ca 250 MPa". This values seems pretty high. In section 4.4 the authors discuss the effects of softening mechanisms - e.g., lower effective friction to mimic the presence of pore fluid-pressure. I was wondering what is level of deviatoric stresses when the model is under hydrostatic conditions.**

Deviatoric stresses for hydrostatic conditions can also be around 200 to 250 MPa (see e.g. Kohlstedt et al., JGR, 1995; their figure 10). However, the maximal stress at the brittle-ductile transition in the cover units is not of first-order importance for our simulations, because the important deformation takes place around the basement-cover contact far below the brittle-ductile transition.

Figure 1: could you please add a small inset to locate the region of the cross-section?

We added a small inset.

I hope my comments contribute the authors to improve the manuscript.
Luca Dal Zilio

---

## Author Comment (AC2) · 10 Jan 2020

The paper presents a set of high-resolution thermo-mechanical simulations aiming to- wards a "nappe theory". The simulations focus strongly on reproducing as many features and attributes documented in the Helvetic nappe system, which guides the choice of input parameters, geometry, and boundary conditions. From a reference simulation, a set of key parameters are varied to test their influence on the simulation outcome.   cˊ First, the viscosity of each material is tested in turn by (i) dropping the viscosity of the basement, (ii) increasing the viscosity of the cover sequence, and (iii) increasing the viscosity of the stronger syn-rift unit capping the rift basins.   cˊ Then the stronger syn-rift unit is replaced by a 4- or 5-layer system involving 2 stronger layers and 2 or 3 weaker units. Three simulations test various thicknesses and configurations.   cˊ Two strain weakening mechanisms (i.e. shear heating and accumulated plastic strain) are tested at various extensional velocities (1 cm/yr and 5 cm/yr). The simulation outcomes are then compared to the Helvetic nappe system.

The paper will be of great interest to geologists interested in nappe tectonics and in particular those interested in the Helvetic nappe system. The paper is well organized, relatively easy to follow, and the figures serve their purpose reasonably well.

The study is an attempt to learn about nappe tectonics from reproducing via numerical simulations the well-documented Helvetic nappe system. However, it remains to be seen whether an all-encompassing "nappe theory" can be extracted from such an approach, for two reasons:   cˊ I would first question in the present context the use of the word "theory". In natural science, a theory is a very robust model established over decades of data collection and analysis and explaining a very large range of unrelated observations. Plate tectonics and biological evolution are two theories. For this reason, I think that the concept of "nappe theory" could safely be replaced by the concept of "nappe model".   cˊ In addition, I think it is pretty safe to state that there is probably more than one way for nappes to develop. Hence, the proposed model is only strictly relevant to the Helvetic nappe system that develops as the result of the inversion of an extended continental margin, and the extrusion of its syn-rift sedimentary infilling. Hence, I think that modifying slightly the title and introduction, to bring a stronger focus on the "Helvetic style" of nappe tectonics, would be beneficial to the paper.

Perhaps the main missing ingredient in the numerical experiments presented here is isostasy and the absence of flexure despite up to 10 km of topography due to crust thickening and nappe stacking. I acknowledge that this issue is touch upon in section 5.2, but it is important to stress that the outcome of this set of simulation will change should the basement be allowed to subside under the weight of the nappe stack.

We added a sentence that the impact of flexure and isostasy on our model results can be tested eventually with larger scale models including flexure and isostasy.

The paper would also benefit from being leaner. I found at places the paper to be unnecessarily wordy, and the description on the simulation lengthy and tedious to read. Rather than describing the evolution of each experiment in great detail (perhaps you can point toward movies or animations instead), it would be best to highlight key differences. The conclusion

needs to be rewritten and shortened. A conclusion goes beyond merely repeating what was said before.

We have added one animation as a supplementary material.

We rewrote and shortened the conclusion and focused on the results relevant for the tectonic interpretation of the Helvetic nappe system.

The supplementary section needs some editing, there are too many spelling mistakes.

We considered and implemented all corrections, from the supplementary pdf. We considered all suggestions and implemented most of them.

Finally, either a code is made freely available, or it is not. Having to ask permission to the author to access the code is, in my view, not sufficient. Codes which are accessible are available online (e.g. underworldcode.org). Chances are that in ten years Underworld will still be available like it was ten years ago.

The code is planned to make publicly available in the near future once the paper describing it is published. The data is not publicly available because of its massive size.

Kind regards, Patrice Rey